# Diagnosis of Cubital Tunnel Syndrome Using Deep Learning on Ultrasonographic Images

**DOI:** 10.3390/diagnostics12030632

**Published:** 2022-03-04

**Authors:** Issei Shinohara, Atsuyuki Inui, Yutaka Mifune, Hanako Nishimoto, Kohei Yamaura, Shintaro Mukohara, Tomoya Yoshikawa, Tatsuo Kato, Takahiro Furukawa, Yuichi Hoshino, Takehiko Matsushita, Ryosuke Kuroda

**Affiliations:** Department of Orthopaedic Surgery, Kobe University Graduate School of Medicine, Kobe 650-0017, Japan; 203m878m@stu.kobe-u.ac.jp (I.S.); mf_ship@yahoo.co.jp (Y.M.); hanako-nishi@live.jp (H.N.); koheidesuyo@yahoo.co.jp (K.Y.); no.8-shintaro@hotmail.co.jp (S.M.); tomo_yoshi_0926@yahoo.co.jp (T.Y.); tkato@med.kobe-u.ac.jp (T.K.); taka1023@med.kobe-u.ac.jp (T.F.); u1ho4no@med.kobe-u.ac.jp (Y.H.); matsushi@med.kobe-u.ac.jp (T.M.); kurodar@med.kobe-u.ac.jp (R.K.)

**Keywords:** artificial intelligence, cubital tunnel syndrome, deep learning, ulnar nerve, ultrasonography

## Abstract

Although electromyography is the routine diagnostic method for cubital tunnel syndrome (CuTS), imaging diagnosis by measuring cross-sectional area (CSA) with ultrasonography (US) has also been attempted in recent years. In this study, deep learning (DL), an artificial intelligence (AI) method, was used on US images, and its diagnostic performance for detecting CuTS was investigated. Elbow images of 30 healthy volunteers and 30 patients diagnosed with CuTS were used. Three thousand US images were prepared per each group to visualize the short axis of the ulnar nerve. Transfer learning was performed on 5000 randomly selected training images using three pre-trained models, and the remaining images were used for testing. The model was evaluated by analyzing a confusion matrix and the area under the receiver operating characteristic curve. Occlusion sensitivity and locally interpretable model-agnostic explanations were used to visualize the features deemed important by the AI. The highest score had an accuracy of 0.90, a precision of 0.86, a recall of 1.00, and an F-measure of 0.92. Visualization results show that the DL models focused on the epineurium of the ulnar nerve and the surrounding soft tissue. The proposed technique enables the accurate prediction of CuTS without the need to measure CSA.

## 1. Introduction

Cubital tunnel syndrome (CuTS) is an entrapment neuropathy caused by the compression and traction of the ulnar nerve at the elbow joint [1]. It is the second most common compressive neuropathy of the upper extremities, with a reported incidence of 25 cases per 100,000 person-years in males and 19 cases per 100,000 person-years in females [2]. Typical CuTS symptoms include numbness, tingling, and dysesthesia in the ring/small finger and dorsum of the hand [3]. As the disease progresses, atrophy of the hypothenar and first dorsal interosseous muscles occurs, which may interfere with daily activities [1]. CuTS is likelier to be at an advanced stage at the time of intervention than carpal tunnel syndrome (CTS) [4]. Therefore, early diagnosis and treatment are important [1]. The first step in diagnosis is obtaining the patient’s medical history, observing the intrinsic musculature of the hand, and evaluating hand sensation [4]. Froment’s and Tinel’s signs at the cubital tunnel are widely used as physical indicators [5]. When CuTS is suspected based on medical history and physical examination, diagnostic techniques, such as magnetic resonance imaging (MRI), electromyography (EMG), and ultrasonography (US), are used [4]. The American Academy of Neuromuscular and Electrodiagnostic Medicine reports the following two measures as the basis for the diagnosis of CuTS: conduction velocity across the elbow is less than 50 m/s, and conduction velocity from above the elbow to below the elbow is slower than that from below the elbow to the wrist by 10 m/s or more [6]. Although EMG testing can help determine the severity of CuTS, MRI can confirm nerve enlargement and compression and is reported to have a 25% higher diagnostic sensitivity than EMG [7]. However, it is impossible to determine the severity of the disease or the prognosis for recovery using MRI [7]. Although both tests are important for diagnosis and staging, they are time consuming and expensive. Furthermore, there are some patients with apparent clinical symptoms of CuTS without abnormal EMG findings, and the diagnostic accuracy (sensitivity 78%) is currently lower than that of the median nerve [8]. US is a minimally invasive and low-cost diagnostic technique that can assess the size of the ulnar nerve. Additionally, US imaging allows for dynamic evaluation, whereas MRI is static. In recent years, high-resolution US has become increasingly effective in the diagnosis of entrapment neuropathies (e.g., CuTS and CTS) [8]. US-based diagnostic studies have shown an enlarged nerve cross-sectional area (CSA) in patients with CuTS symptoms, even when the EMG is normal [8]. The measurement of CSA by US imaging is expected to provide an early diagnosis of CuTS. Chang et al. reported that when CSA > 10 mm^2^, the diagnostic accuracy is high, with a sensitivity of 0.85 and a specificity of 0.91 [9,10]. Although there is room for further study on CSA thresholds, US imaging is becoming the first choice for the imaging of peripheral lesions of limbs [11]. On the other hand, US imaging has limitations, such as anisotropy and a lack of technical proficiency [12]. These factors invariably cause errors in CSA measurement, which, in turn, diminish diagnostic accuracy. Therefore, this study focuses on the application of DL to US images.

In recent years, there has been remarkable progress in the field of computer science, due to the increase in processing speed, which has become increasingly popular in medical image processing [13]. In the field of medical imaging, deep learning (DL) using convolutional neural networks (CNNs) is widely used to automatically learn image features [14]. These methods are now being used for various applications, such as the diagnosis of pneumonia and fractures in X-rays [15,16]. DL diagnostic methods have also been applied to US imaging, and its diagnostic accuracy for thyroid nodules is 83% [17]. Additionally, DL visualization methods can intuitively identify important areas to further examine for disease prediction [18]. In the field of orthopedics, DL-based diagnostic imaging techniques have been attracting attention in recent years and the numbers of reports are gradually increasing. Throughout the orthopedic literature, DL has been used for diagnostic purposes in the study of hip and knee joints based on medical imaging [19]. In the diagnosis of osteoarthritis (OA) of the hip joint using X-rays, the CNNs model achieves an accuracy of 92.8%, which is comparable to that of an attending physician with 10 years of experience in diagnosing hip OA [20]. For the knee joint, Ashinsky et al. used the DL algorithm for T2-weighted maps of the medial femoral epicondyle to predict progression to clinically symptomatic OA; the DL algorithm was able to predict the development of OA with 75% accuracy [21]. As shown in these results, DL is reported to have promising potential in the field of intelligent medical imaging practice, especially for OA of the hip and knee joints. On the other hand, there has been no study using DL for entrapment neuropathies, which is becoming the main diagnostic tool using US images. If AI technology can predict diseases as accurately as conventional diagnostic methods, its application in clinical practice may lead to early diagnosis. Therefore, the purpose of this study is to calculate diagnostic accuracy from a confusion matrix using DL on US CuTS images. The hypothesis of this study is twofold: the application of DL to US imaging allows CuTS diagnosis without the need to measure CSA, and it does so at an accuracy at least as high as conventional MRI and EMG methods. Furthermore, we considered the possibility of obtaining useful information for the diagnosis of CuTS by visualizing areas deemed important. To the best of our knowledge, this study is the first to investigate the use of DL on US images for CuTS diagnosis.

## 2. Materials and Methods

The ethics committee in our hospital approved this study (No. B21009), and informed consent was obtained from all participants.

We collected elbow US images from 60 participants—30 healthy volunteers comprising 16 males and 14 females (control group), and 30 patients comprising 18 males and 12 females who were diagnosed with CuTS via EMG (patient group). The criteria for diagnosis by EMG were based on data from previous reports [22]: absolute motor nerve conduction velocity (mNCV) of less than 50 m/s across the elbow and a decrease in velocity of more than 10 m/s above and below the elbow. EMG of the patient group showed a decrease in mNCV (≤50 m/s) across the elbow and a decrease in velocity of more than 10 m/s above and below the elbow in all subjects. A decrease in the amplitude of the compound muscle action potential was observed in 13 patients. Patients with a history of surgery on the elbow, including surgery for CuTS, were excluded. The mean ages of the control and patient groups were 55.6 ± 11.5 years (range: 28–75) and 63.8 ± 13.2 years (range: 43–75), respectively. The sample size was determined by performing a statistical power analysis, based on data from a previous study, using G*Power 3.1 software [10]. A prior sample size calculation performed at the start of this study showed that a difference in CSA of 3 mm^2^ was detectable in two groups, with a sample size of 26 participants (13 in each group), using a t-test (effect size = 1.2, α error = 0.05, power = 0.9). US imaging was performed by one certified hand surgeon (A.I.) with 11 years of musculoskeletal ultrasound imaging experience. US imaging showed a short-axis image of the ulnar nerve at the level of the medial epicondyle using an 18M linear probe (Figure 1: Canon APLIO300, TUS-A300, Canon Medical Systems, Tochigi, Japan). The US settings of gain, dynamic range, and frame rate were standardized for all measurements. A US movie was captured by sliding the probe within a 30 mm range around the medial epicondyle. From the obtained US movies, 100 images were captured per elbow, and 3000 images were prepared from each group. An area of 25 × 25 mm, that included the ulnar nerve, was cropped per image prior to DL processing. For DL, the DeepLearning Toolbox in MATLAB (Mathworks, Natick, MA, USA) [23] was used. The architecture of the framework is shown in Figure 2.

### 2.1. Data Selection

Of the 30 participants in each group, data from 25 participants were randomly selected to validate training by 5-fold cross validation. The images from the remaining five participants were used as test data to evaluate the model performance.

### 2.2. Data Augmentation

For preprocessing, data augmentation was performed to increase the variation in the original dataset. The *ImageAugmentor* tool in MATLAB was used to augment training and validation images by applying horizontal flipping, rotation (−10° to 10°), scaling (×0.8 to ×1.2), horizontal translation, vertical translation, and random shearing.

### 2.3. Training Process

A total of 2500 images were randomly selected from each group and used as training data (Figure 3); the remainder were used as testing data. Transfer learning was performed using three pre-trained models (i.e., Residual Net (ResNet)-50, MobileNet_v2, and EfficientNet). These models used different numbers of convolutional layers: 50 in ResNet-50, 53 in MobileNet_v2, and 82 in EfficientNet. The block diagram of each learning model is shown in Figure 4, Figure 5 and Figure 6. Patient data were randomly divided into two groups (training and testing). During training, five-fold cross validation was performed [24]. In this procedure, training data were randomly divided into five subsets: one was used for validation, and the remaining four were used for training. This process was repeated 5 times until each subset was used exactly once for validation. The hyperparameters of the training models were determined by using the Experimental Manager application in the DeepLearning Toolbox and the Parallel Computing Toolbox. The parameters are summarized in Table 1. To avoid overfitting, training and validation images were augmented by employing the random transformation process defined earlier. The model also validated the network, every 50 iterations, by predicting the responses of the validation data and calculating the loss and accuracy.

The accuracy of each DL model was evaluated using a confusion matrix, which is a table containing four combinations of predicted and actual values regarding the presence or absence of disease. The matrix values include true positives, false positives, true negatives, and false negatives (Figure 7).

### 2.4. Visualization of Important Features

Local interpretable model-agnostic explanations (LIMEs) and occlusion sensitivities were used to visualize image features deemed important by the DL model [18].

### 2.5. Measures

The accuracy of the three learning models was evaluated using testing data. The evaluation terms were defined as follows: accuracy (percentage of correct answers for all data), precision (percentage of the patient group correctly judged), recall (percentage of data correctly judged as the patient group, same as sensitivity), specificity (percentage of data correctly judged as the control group), and F-measure (the harmonic mean of accuracy and recall). Additionally, the accuracy of the classifier was evaluated based on the area under the receiver operating characteristic (ROC) curve (AUC). The 95% confidence interval (CI) for each endpoint was calculated using the bootstrap method [25], where 500 replacement resampling procedures were performed to evaluate the model performance on test data. To verify the absence of overfitting training data, all data were validated by five-fold cross validation [24]. For statistical analysis, a one-way ANOVA was performed using IBM SPSS Statistics v.21 (IBM, Armonk, NY, USA).

## 3. Results

The AUCs for CuTS prediction using the test data with ResNet-50, MobileNet_v2, and EfficientNet were 0.987 (0.984–0.989), 0.990 (0.987–0.991), and 0.989 (0.987–0.991), respectively (Figure 8). The high AUC indicates the high accuracy of the classifier. For each learning model, the prediction accuracy based on the confusion matrix obtained from the test data is shown in Table 2. The confusion matrix for each learning model is shown in Figure 9. The results of cross validation show no statistically significant differences among the learning models or between the scores of validation and testing, suggesting that overfitting did not occur.

The visualization of important features, using image LIMEs and occlusion sensitivity, verified that all three models predicted the presence of CuTS by focusing on hyperechoic changes in the ulnar nerve epithelium and those in the ulnar nerve interior and surrounding tissue (Figure 10).

## 4. Discussion

Applying DL to US CuTS images resulted in highly accurate lesion predictions. The best scores were 0.90 for accuracy (ResNet-50, MobileNet_v2), 0.86 for precision (ResNet-50, MobileNet_v2), 1.00 for recall (all models), and 0.92 for F-measure (ResNet-50, MobileNet_v2), indicating that diagnoses may be determined with high accuracy, even without CSA measurement. Therefore, the proposed technique can be useful in screening for CuTS. As mentioned, this study is the first to investigate the use of DL on US images for CuTS diagnosis.

In recent years, US imaging has been widely used for the diagnosis of musculoskeletal disorders [26,27]. In particular, the use of US imaging for the diagnosis of entrapment neuropathy is rapidly gaining popularity. Nerve enlargement has been observed centrally above the nerve strangulation site, and the measurement of CSA is helpful in diagnoses [28]. The measurement of the median nerve CSA can be used to diagnose CTS with high accuracy (sensitivity 83%; specificity 86%) [29]. Furthermore, the measurement of the ulnar nerve CSA via US is also useful in the diagnosis of CuTS [9,10]. The ulnar nerve CSA in healthy volunteers rarely exceeds 10 mm^2^ at various sites; thus, it is used as the cut-off value for CuTS [10]. Although US imaging is useful in diagnosing CuTS, measuring CSA by identifying and drawing the entrance is challenging. Therefore, we focused on the advantages of DL, which automatically learns image features and has been applied extensively to the analysis of medical data [15]. The application of DL to diagnostic imaging has been actively performed since 2015 [30]. Bar et al. evaluated the diagnostic accuracy of DL by classifying chest radiographs into healthy and pathological (e.g., cardiomegaly, right pleural effusion, and multiple others). A sensitivity of 0.89 and a specificity of 0.79 were observed in their study, indicating that DL using non-medical image databases can be used for general medical image recognition tasks [30]. Diagnostic imaging technology using DL has also been applied to orthopedics [31,32]. Amelia et al. applied DL to images of proximal femoral fractures and reported a prediction accuracy of 0.94 [31]. Additionally, Tomita et al. applied DL to computer tomography (CT) images and reported a diagnostic accuracy of 0.89 for osteoporotic vertebral fractures [33]. DL has also been used with US images to identify the vasculature in the axillary approach of the brachial plexus [34]. The results show that CNNs have a recall of 0.81 and a precision of 0.88 in detecting blood vessels [34]. Although imaging diagnoses using DL have been gradually reported in the orthopedic research field, there are no reports regarding the use of DL in diagnosing entrapment neuropathy. Research on DL for US images is ongoing in the field of breast surgery. The mortality rate of breast cancer is high, and DL has been studied heavily in its US diagnosis. Hence, diagnostic accuracy improvements have been at the forefront of research. Jabeen et al. obtained the best CNN model accuracy to date, of 99.1%, by optimizing five subprocesses: data augmentation, modified DarkNet-53 modeling, transfer learning, feature selection, and feature fusion and classification [35]. Feature fusion augments the features of small lesions by combining feature maps of shallow and deep convolutional layers and is commonly applied to mammography [36]. Meraj et al. used a quantization-assisted U-Net with data augmentation to accurately extract lesions from ultrasound images [37]. Although the lesions are sufficiently segmented by U-Net, the results show that the use of quantization for some segmented lesions leads to increased accuracy (≥96%) [37].

Early detection of entrapment neuropathy is also desirable as the progression of symptoms can interfere with daily life. In this field, the diagnosis of entrapment neuropathy by CNNs has not been reported. Since this is the first application of DL to US images of CuTS, this study focuses on comparing the diagnostic accuracy, using three popular pretrained DL models, and visualizing the basis for decision. In the present study, the possibility of predicting CuTS without measuring CSA was investigated by applying CNNs, which captured and evaluated the image features of CuTS diagnoses. The results show that the DL models used in this study predicted CuTS with high accuracy. As indicated by the confusion matrix results, no DL model mistakenly judged CuTS in the control group. US is minimally invasive, and the study shows the possibility of using the DL models for CuTS screening. Compared with previous reports that measured CSA (sensitivity 0.85, specificity 0.91) [9], each learning model in this study was particularly sensitive in predicting CuTS and was effective for screening. The extension and clinical application of this system are expected to facilitate the early detection of CuTS.

Notably, three DL models that are popularly used for medical image classification were selected for this study. ResNet50 solves the accuracy loss of DL by reconfiguring it to learn residual functions [38] to reduce the error rate and achieve high accuracy. ResNet50 has been used with MRI images of ACL injuries [39] and other US images [40]. MobileNet_v2 simplifies DL by introducing inversion residuals with linear bottlenecks which improve efficiency and reduce the memory footprint [41]. MobileNet_v2 is efficient in image classification and object detection and it can achieve the same, or higher, accuracy as the same parameter model with high speed. It has been applied to lung CT [42]. EfficientNet is one of the most powerful CNN architectures reported in recent years [43]. It utilizes a complex scaling method to increase the depth, width, and resolution of the network, and provides state-of-the-art capabilities with fewer computational resources than other models [43]. EfficientNet can process image data 6.1 times faster than ResNet, using 8.4 times fewer memory resources [44]. Although memory cost and processing speed are vital to DL-based diagnostic imaging, the prediction accuracy of ResNet and MobileNet_v2 was higher than that of EfficientNet in this study. EfficientNet emphasizes its deep convolutional layers and strong learning capability; this may lead to the problem of overlearning, as it retains too much information about each pixel in an image. The development of CNNs has been remarkable, and new models are being developed every year, but it may be important in the future to optimize an appropriate learning model for each diagnostic modality.

Although accuracy is essential in AI-based diagnostic imaging technology, the basis for model decision making must be understandable to humans. In particular, DL was used in this study for diseases lacking a US diagnosis basis, and emphasis was placed on visualizing DL decision making rather than on depicting small lesions. Models falling within the purview of explainable AI include gradient-weighted class activation mapping (Grad-CAM), occlusion sensitivity, and LIME [18]. Grad-CAM visualizes important pixels by weighting the gradient against the predicted value, which is useful for image classification [45]. Grad-CAM was not used in this study owing to its low spatial resolution when assessing the presence of a disease. In the present study, CuTS prediction was based on a detailed comparison with a control group using occlusion sensitivity and LIME. Occlusion sensitivity can effectively visualize multifocal glass opacities and consolidations, allowing detailed visualizations of important image features [23], whereas LIME extracts important regions by creating superpixels and has attracted significant attention in the field of medical imaging [46]. These visualization tools may provide the necessary information for diagnosis by highlighting the basis for predictions made by the DL model. In this study, the DL model focused on hyperechoic changes in the ulnar nerve epithelium, and the hypoechoic changes in the ulnar nerve interior and surrounding tissue, for feature visualization. The model successfully recognized the nerve swelling and edema of the surrounding tissue from US images. Thus, it is possible to diagnose CuTS with high accuracy without measuring CSA. Furthermore, because DL models learn image features using an uninhibited and unbiased neural model compared with humans, DL feature visualizations may enable physicians to detect previously overlooked and unquantified features. Early CuTS diagnosis is important as delays can lead to muscle weakness and atrophy. The clinical application of DL for US imaging may lead to early CuTS diagnosis, and this system is expected to expand in the future.

This study has the following limitations. First, the results were obtained exclusively with US images, and the generalization of the performance of the models on images from other diagnostic devices was not investigated. Second, no accuracy comparison with other diagnostic devices was performed. Moreover, although many images were reviewed, the number of cases considered was not large. Further research is required to corroborate the results of this study. Finally, in this study, the same US device was used in every phase, and no investigation was conducted regarding changes in the environment, such as the addition of noise. The same image may not be obtained even when looking at the same object because the settings of the US device and the examiner’s skill may vary. Although multiple patients were examined experimentally using the same US device from different angles by tilting, sliding, and rotating the probe under the same gain and focus conditions, the accuracy of the measurements should be verified by adding data obtained from different instruments to improve generalizability.

## 5. Conclusions

In this study, US images of CuTS regions of interest were evaluated using AI-based medical imaging technology. US images using the DL model predicted CuTS with high accuracy, with the best accuracy score of 0.90 for ResNet-50 and MobileNet_v2, the best precision score of 0.86 for ResNet-50, the best sensitivity score of 1.00 for all models, and the best F-measure score of 0.92 for ResNet-50 and MobileNet_v2. Furthermore, all models successfully predicted CuTS by focusing on the neural interior and perineural tissue of the ulnar nerve. The proposed method can be improved in the future to provide a more convenient and accurate CuTS diagnosis using US imaging, which is minimally invasive, low cost, and excellent for soft tissue examinations. We expect DL-based systems to make the diagnosis of CuTS easier and more accurate.

## Figures and Tables

**Figure 1 diagnostics-12-00632-f001:**
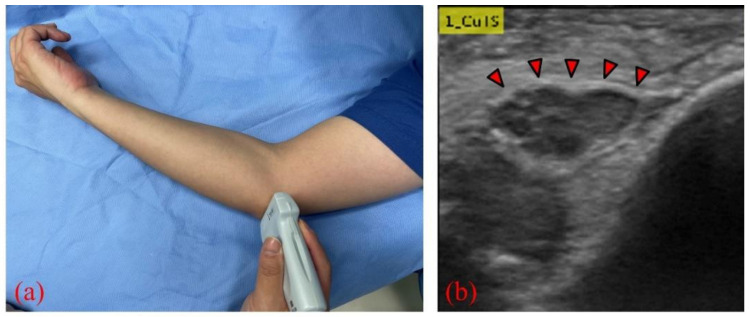
(**a**) US probe placed on the medial epicondyle to visualize the ulnar nerve; (**b**) short-axis image of the ulnar nerve (red arrows) at the level of the medial epicondyle.

**Figure 2 diagnostics-12-00632-f002:**
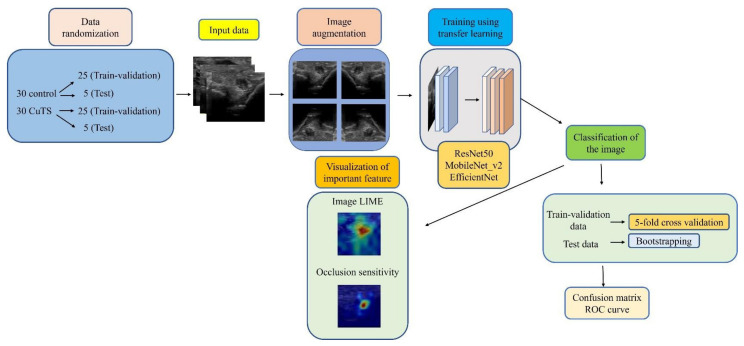
Flowchart of the proposed framework.

**Figure 3 diagnostics-12-00632-f003:**
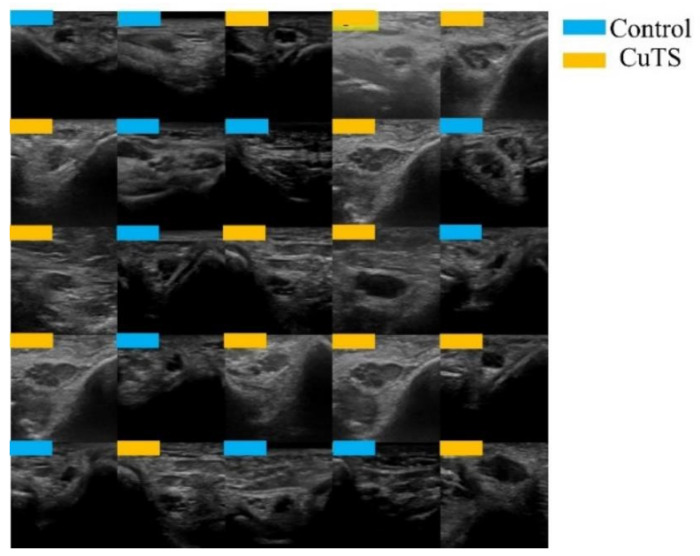
Images were randomly extracted by AI to be used as training data (light blue for control, orange for CuTS patients).

**Figure 4 diagnostics-12-00632-f004:**
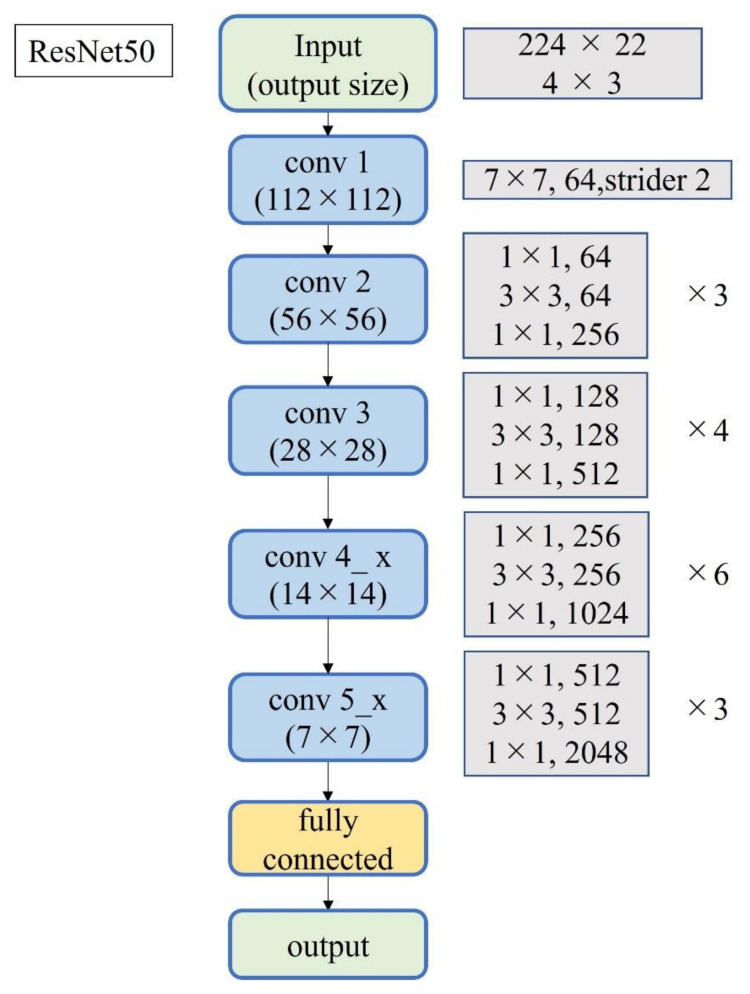
Block diagram of ResNet-50.

**Figure 5 diagnostics-12-00632-f005:**
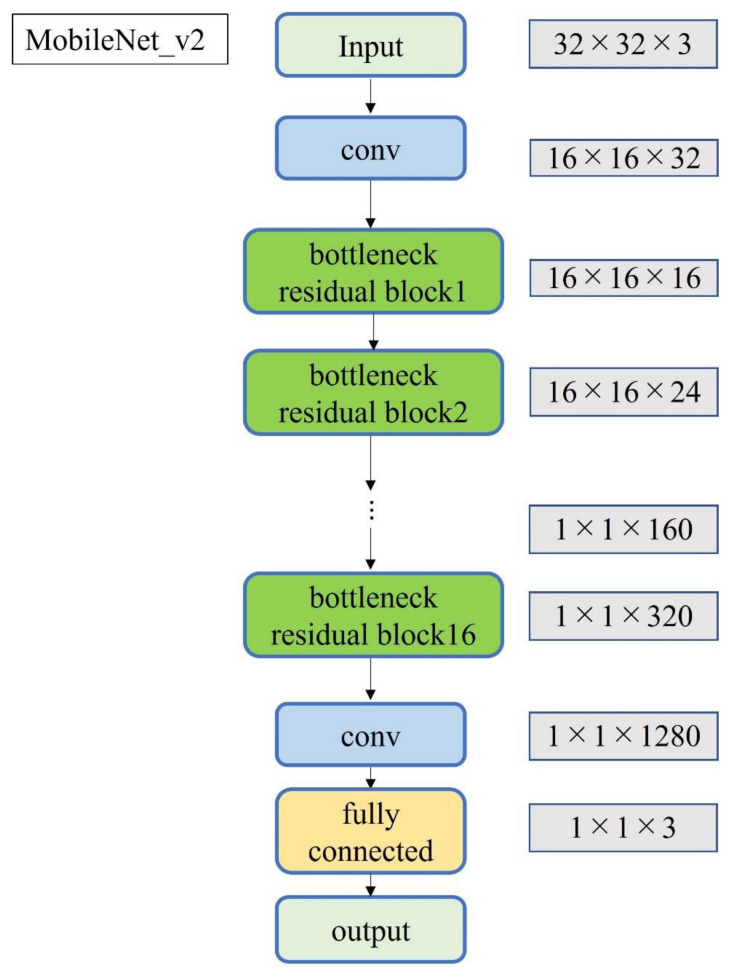
Block diagram of MobileNet_v2.

**Figure 6 diagnostics-12-00632-f006:**
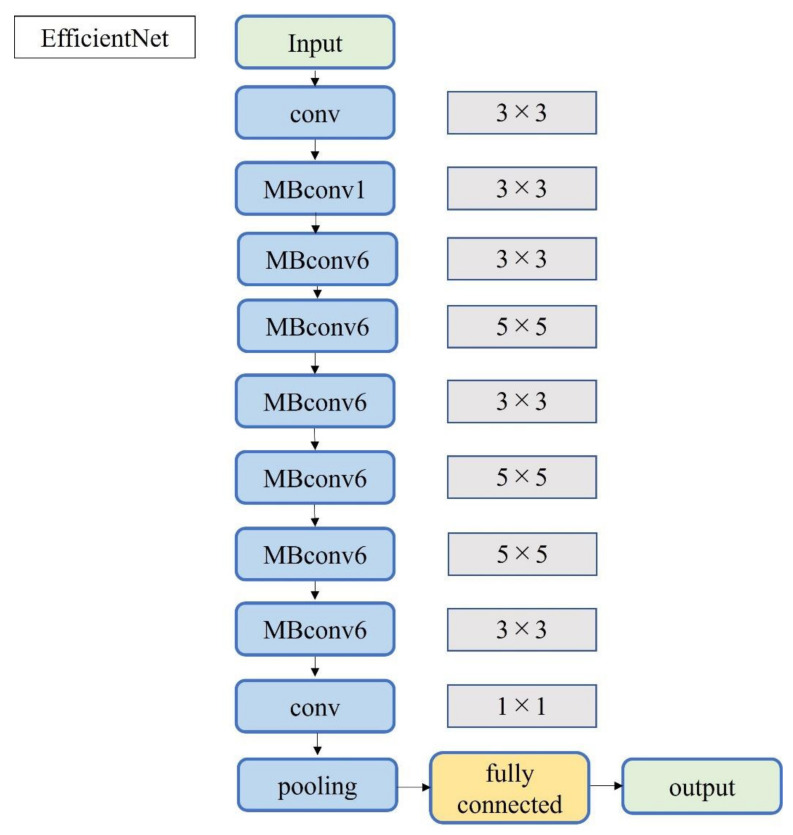
Block diagram of EfficientNet.

**Figure 7 diagnostics-12-00632-f007:**
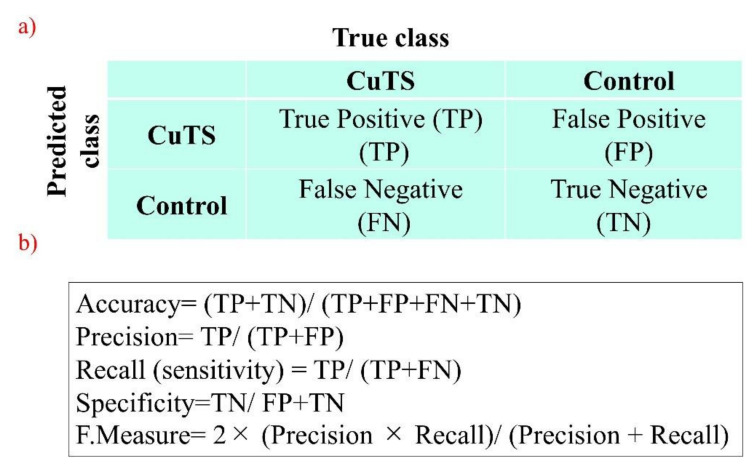
(**a**) A confusion matrix is a table of four combinations based on predicted and actual values and the presence or absence of disease; (**b**) diagnostic accuracy from the learning model is calculated from the confusion matrix created using testing data.

**Figure 8 diagnostics-12-00632-f008:**
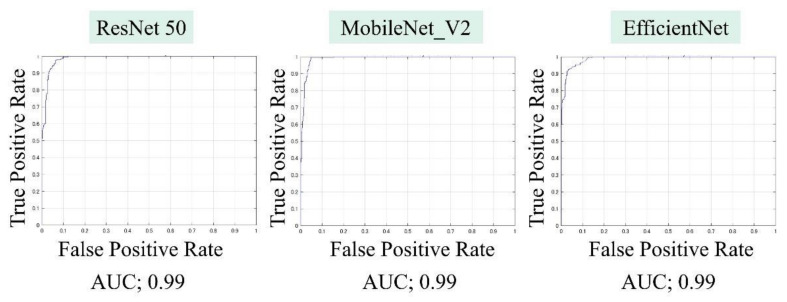
Area under the curve (AUC), based on the receiver operating characteristic (ROC) curve, was high for all learning models.

**Figure 9 diagnostics-12-00632-f009:**
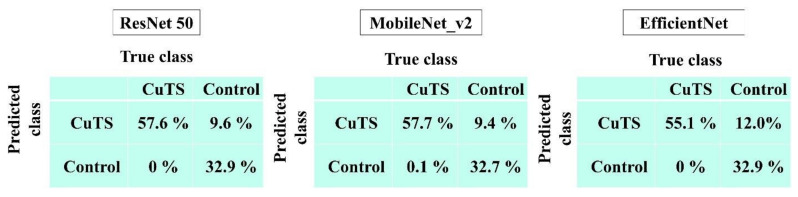
Confusion matrix of each learning model.

**Figure 10 diagnostics-12-00632-f010:**
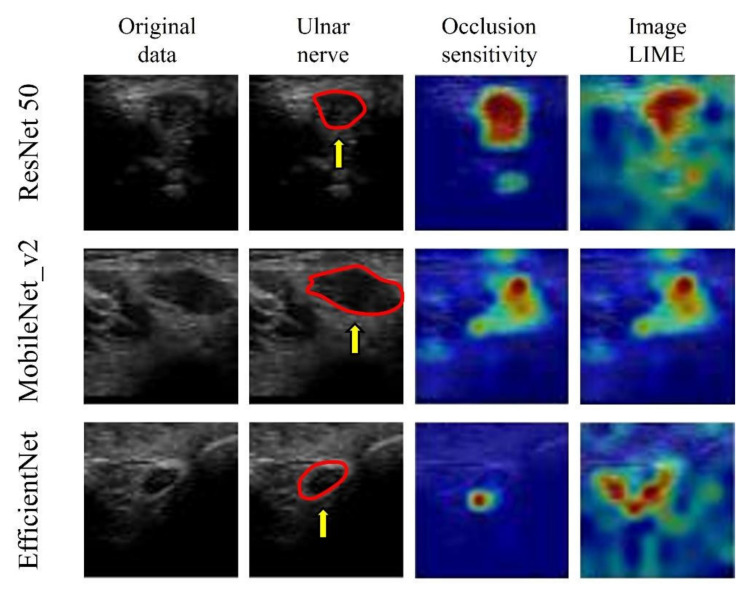
Visualization of the region of interest using occlusion sensitivity and image LIMEs. Learning models focus on neural interior and perineural tissues. The red circle is a cross section of the ulnar nerve in the original image. AI focused on hyperechoic changes in the ulnar nerve epithelium and hypoechoic changes in the ulnar nerve interior and surrounding tissue.

**Table 1 diagnostics-12-00632-t001:** Best parameters of the training model.

	ResNet-50	MobileNet_v2	EfficientNet
Optimizer	Adam *	Adam	Adam
MiniBatchsize	20	10	10
Epochs	1000	700	1000
Learning rate	0.0001	0.0001	0.0001

* Adam; adaptive moment estimation.

**Table 2 diagnostics-12-00632-t002:** Diagnostic accuracy from the learning model was calculated from the confusion matrix created from testing data. The best accuracy score was 0.90 in ResNet-50 and MobileNet_v2; precision was 0.86 in ResNet-50 and MobileNet_v2; recall was 1.0 in all models; and the F-measure was 0.92 in ResNet-50 and MobileNet_v2.

Network	Accuracy	Precision	Recall (Sensitivity)	Specificity	F-Measure
ResNet-50	0.904 (0.903–0.907)	0.859 (0.856–0.861)	1.00	0.774 (0.772–0.776)	0.924 (0.922–0.925)
MobileNet_v2	0.904 (0.902–0.906)	0.859 (0.857–0.862)	0.998 (0.997–0.998)	0.776 (0.774–0.778)	0.923 (0.921–0.925)
EfficientNet	0.880 (0.878–0.882)	0.821 (0.819–0.824)	1.00	0.732 (0.731–0.734)	0.902 (0.900–0.904)

95% confidence interval.

## Data Availability

The data presented in this study are available on request from the corresponding author. The data are not publicly available due to confidentiality concerns.

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
