# Peer review of "Diagnosis of Cubital Tunnel Syndrome Using Deep Learning on Ultrasonographic Images"

_diagnostics, 2022, doi:10.3390/diagnostics12030632_

Round 1
Reviewer 1 Report
Authors evaluated Cubital tunnel syndrome using ultrasound techniques with deep learning techniques. They showed some improved recall data. Authors mentioned that they first investigate the use of DL on US images in CuTS diagnosis. However, I am wondering which new techniques of deep learning-based US are used to improve the recall data. In addition, authors must ask native English colleague professor or professional services because of some broken English expression. Authors also need to describe in detail about the concept, procedures, and some images. Therefore, the manuscript should be improved with some suggestive opinions as below.
1. Please correct 2500 to 2,500 in Line 14.
2. In Figure 2, fonts are not clear to be seen.
3. Figure 4 font quality is so low and font sizes are too small.
4. Authors need to specify the detail procedures and block diagrams of the deep learning techniques. Otherwise, this manuscript could be case report.
5. In Figure 5, authors need to mark important areas in the images to be visualized.
6. In Table 1, some data does not show some improvement. Thus, authors can specify the opinion.
7. Authors need to describe what is ResNet 50, MobileNet, EfficientNet because the manuscript need to be open to readers who are not familiar with US and AI research areas.
Author Response
Thank you very much for reviewing our manuscript. Please see the attachment. We would like to thank you for your cooperation.
Response to the Reviewer 1
Authors evaluated Cubital tunnel syndrome using ultrasound techniques with deep learning techniques. They showed some improved recall data. Authors mentioned that they first investigate the use of DL on US images in CuTS diagnosis. However, I am wondering which new techniques of deep learning-based US are used to improve the recall data. In addition, authors must ask native English colleague professor or professional services because of some broken English expression. Authors also need to describe in detail about the concept, procedures, and some images. Therefore, the manuscript should be improved with some suggestive opinions as below.
- Response: We thank the reviewer for their pertinent comments. As far as possible, we have modified the manuscript to incorporate your critical suggestions. We have tried to retain the basic format and arguments of the previously submitted paper. We also had our English proofreading partner check the manuscript. The key changes in the manuscript are highlighted in yellow.
- Please correct 2500 to 2,500 in Line 14.
- Response: Thank you for highlighting this. We have made the corrections as you suggested (Page 1, line 14; Page 3, lines 119 and 133).
In Figure 2, fonts are not clear to be seen. - Response: Thank you for pointing this out. The font size of the figures (Figures 3, 7) has been corrected.
- Figure 4 font quality is so low and font sizes are too small.
- Response: Thank you for pointing this out. The font size of Figure 4 (new Figure 8) has been corrected.
Authors need to specify the detail procedures and block diagrams of the deep learning techniques. Otherwise, this manuscript could be case report. - Response: Thank you for pointing this out. Detailed procedures and block diagrams of deep learning techniques have been added. The detailed procedure has been shown in Figure 2, and block diagrams have been added as Figures 4–6.
- In Figure 5, authors need to mark important areas in the images to be visualized.
- Response: Thank you for pointing this out. The contour of the short axis of the ulnar nerve at the cubital tunnel in the original image has been added (Figure 10).
- In Table 1, some data does not show some improvement. Thus, authors can specify the opinion.
- Response: The numbers in parentheses are not comparisons, but are values with 95% confidence intervals. To avoid misunderstanding, we have added a note below the table. The results were also evaluated by using the cross-validation method. No significant differences were found for any factor. Relevant information has been added to the manuscript (Page 8, lines 196–198). The revised text is as follows:
The results of cross-validation showed no statistically significant differences among the learning models or between the scores of validation and testing, suggesting that overfitting did not occur.
7. Authors need to describe what is ResNet 50, MobileNet, EfficientNet because the manuscript need to be open to readers who are not familiar with US and AI research areas.
- Response: Thank you for your suggestion. A brief summary has been provided for each learning model (Pages 10– 11, lines 278–289). The revised text is as follows:
ResNet50 solves the accuracy loss of DL by reconfiguring it to learn residual functions [38] to reduce the error rate and achieve high accuracy. ResNet50 has been used with MRI images of ACL injuries [39] and other US images [40]. MobileNet_v2 simplifies DL by introducing inversion residuals with linear bottlenecks, which improve efficiency and reduce the memory footprint [41]. MobileNet_v2 is efficient in image classification and object detection, and it can achieve the same or higher accuracy as the same parameter model with high speed. It has been applied to lung CT [42]. EfficientNet is one of the most powerful CNN architectures reported in recent years [43]. It utilizes a complex scaling method to increase the depth, width, and resolution of the network and provides state-of-the-art capabilities with fewer computational resources than other models [43]. EfficientNet can process image data 6.1 times faster than ResNet, using 8.4 times fewer memory resources [44].
Reference
[38] Kaiming He; Xiangyu Zhang; Shaoqing Ren; Sun, J. Deep Residual Learning for Image Recognition. arXiv: 1512.03385 2015.
[39] Zhang, L.; Li, M.; Zhou, Y.; Lu, G.; Zhou, Q. Deep Learning Approach for Anterior Cruciate Ligament Lesion Detection: Evaluation of Diagnostic Performance Using Arthroscopy as the Reference Standard. J Magn Reson Imaging 2020, 52, 1745-1752, doi:10.1002/jmri.27266.
[40] Zhang, H.; Han, L.; Chen, K.; Peng, Y.; Lin, J. Diagnostic Efficiency of the Breast Ultrasound Computer-Aided Prediction Model Based on Convolutional Neural Network in Breast Cancer. J Digit Imaging 2020, 33, 1218-1223, doi:10.1007/s10278-020-00357-7.
[41] Mark Sandler; Andrew Howard; Menglong Zhu; Andrey Zhmoginov; Chen, L.-C. MobileNetV2: Inverted Residuals and Linear Bottlenecks. arXiv: 1801.04381 2019.
[42] Gang, L.; Haixuan, Z.; Linning, E.; Ling, Z.; Yu, L.; Juming, Z. Recognition of honeycomb lung in CT images based on improved MobileNet model. Med Phys 2021, 48, 4304-4315, doi:10.1002/mp.14873.
[43] Wang, J.; Liu, Q.; Xie, H.; Yang, Z.; Zhou, H. Boosted EfficientNet: Detection of Lymph Node Metastases in Breast Cancer Using Convolutional Neural Networks. Cancers (Basel) 2021, 13, doi:10.3390/cancers13040661.
[44] Tan, M.; Le, Q.V. EfficientNet: Rethinking model scaling for convolutional neural networks. Proceedings of the 36th International Conference on Machine Learning, 2019, 6105-6114

Reviewer 2 Report
The paper proposes an explainable deep learning methodology for ultrasonic image analysis. The proposed methodology is used for the recognition of cubital tunnel syndrome, while the experimental results are good. The article has some methodological and presentation issues and needs to be revised according to the comments presented below before it could be considered for publication.
Comments:
- The paper is rather short for a journal article (just 8 pages). The paper should be extended in all aspects.
- What is the knowledge gap covered by this study? Convolutional neural networks are used in multiple studies on biomedical image classification and disease recognition with various modifications and optimizations. This paper uses some of the well-known and widely-used deep learning models (ResNet50, MobileNet_v2, and EfficientNet). The novelty of this study and its difference from the previous works must be explicitly stated.
- The paper lacks a separate related works section. The Introduction section discusses (in very general terms) some studies and papers but without much detail. More recent articles should be included in the discussion in this rapidly developing field of research. Discuss various modalities used for breast cancer recognition (X-ray, ultrasound, thermal images, etc.). I suggest to focus on state-of-the-art research works published in high impact journals such as * Breast cancer classification from ultrasound images using Probability‐Based optimal deep learning feature fusion. * A quantization assisted U-net study with ICA and deep features fusion for breast cancer identification using ultrasonic data. * Breast cancer detection using mammogram images with improved multi-fractal dimension approach and feature fusion. Summarize the discussed works in a table. Discuss the limitations of these and other studies as a motivation of your work.
- Present the full process of your methodology as a workflow diagram.
- More information about image pre-processing should be given. Did you use any image augmentation methods?
- Explain the training process of your neural network models. What is the training/test split ratio? What are the values of network hyperparameters for training (learning rate, batch size, etc.)? The work also needs a very detailed presentation of how the optimum parameters were found for network hyperparameters, and how any overfitting was compensated for.
- Line 133: how many samples were used for bootstrapping?
- Add and discuss more experimental results: confusion matrices.
- Table 1: is the difference between the performance of models statistically significant. Perform statistical testing and present the results (p-values).
- The noise in real-world biomedical images is a well-known problem that reduces the accuracy of diagnostics. You should explore the robustness of the proposed method to noise. You can artificially lower the quality for example by adding noise, and then analyse the drop in performance.
- For multi-part figures, explain each subfigure in the caption of the figures.
Author Response
Thank you very much for reviewing our manuscript. Please see the attachment file. Thank you very much for your cooperation.
Response to the Reviewer 2
The paper proposes an explainable deep learning methodology for ultrasonic image analysis. The proposed methodology is used for the recognition of cubital tunnel syndrome, while the experimental results are good. The article has some methodological and presentation issues and needs to be revised according to the comments presented below before it could be considered for publication.
- Response: Thank you for reviewing our manuscript. We are confident that the manuscript will be improved, thanks to your suggestions. The main ideas of the paper have been retained, even after revisions, to address your concerns. Please see our point-by-point responses below.
Comments:
1.The paper is rather short for a journal article (just 8 pages). The paper should be extended in all aspects.
- Response: Thank you for highlighting this. We have further explained the learning model and methodology.
2.What is the knowledge gap covered by this study? Convolutional neural networks are used in multiple studies on biomedical image classification and disease recognition with various modifications and optimizations. This paper uses some of the well-known and widely-used deep learning models (ResNet50, MobileNet_v2, and EfficientNet). The novelty of this study and its difference from the previous works must be explicitly stated.
- Response: Thank you for your question. In this study, the network and algorithm are not new, but we emphasize the possibility of detecting entrapment peripheral nerve diseases (CuTS) by using DL. The diagnosis of peripheral nerve lesions by using ultrasound is not widely accepted, and a clinically accepted method has not yet been established. Delayed detection of CuTS often leads to muscle atrophy and interferes with daily life. Currently, DL-assisted diagnosis is being used in the clinical filed but not in musculoskeletal imaging. The novelty of the current study is the adaptation of DL approaches to US imaging of peripheral nerve diseases.
3.The paper lacks a separate related works section. The Introduction section discusses (in very general terms) some studies and papers but without much detail. More recent articles should be included in the discussion in this rapidly developing field of research. Discuss various modalities used for breast cancer recognition (X-ray, ultrasound, thermal images, etc.). I suggest to focus on state-of-the-art research works published in high impact journals such as * Breast cancer classification from ultrasound images using Probability‐Based optimal deep learning feature fusion. * A quantization assisted U-net study with ICA and deep features fusion for breast cancer identification using ultrasonic data. * Breast cancer detection using mammogram images with improved multi-fractal dimension approach and feature fusion. Summarize the discussed works in a table. Discuss the limitations of these and other studies as a motivation of your work.
- Response: Thank you for your suggestion. We have added the recommended literature review in the discussion section and included the relevant references. In the field of peripheral nerve diseases, electrophysiological testing is the main method of diagnosis, and diagnosis by US imaging is not widely used. According to your recommendation, we have added these points to the discussion section (Page; 10, lines 250–267).
4.Present the full process of your methodology as a workflow diagram.
- Response: Thank you for your suggestion. A flowchart has been added to the Methodology section as Figure 2.
5.More information about image pre-processing should be given. Did you use any image augmentation methods?
- Response: Thank you for pointing this out. Information on image augmentation has been added to the Methodology section (Page 3, lines 128–131). The details added to the manuscript are as follows:
For preprocessing, data augmentation was performed to increase the variation in the original dataset. The ImageAugmentor tool in MATLAB was used to augment training and validation images by applying horizontal flipping, rotation (-10° to 10°), scaling ( × 0.8– × 1.2), horizontal translation, vertical translation, and random shearing.
6.Explain the training process of your neural network models. What is the training/test split ratio? What are the values of network hyperparameters for training (learning rate, batch size, etc.)? The work also needs a very detailed presentation of how the optimum parameters were found for network hyperparameters, and how any overfitting was compensated for.
- Response: Thank you for pointing this out. Detailed information has been added to the Methodology section (Pages 3–4, lines 138–148). The details are as follows:
Patient data were randomly divided into two groups (training and testing). During training, a five-fold cross validation was performed [24]. In this procedure, training data were randomly divided into five subsets: one was used for validation, and the remaining four were used for training. This process was repeated 5 times until each subset was used exactly once for validation. The hyperparameters of the training models were determined by using the Experimental Manager application in the DeepLearning Toolbox and the Parallel Computing Toolbox. The parameters are summarized in Table 1. To avoid overfitting, training and validation images were augmented by employing the random transformation process defined earlier. The model also validated the network every 50 iterations by predicting the responses of the validation data and calculating the loss and accuracy.
Reference
[24] Srinivas, S. A Machine Learning-Based Approach for Predicting Patient Punctuality in Ambulatory Care Centers. Int J Environ Res Public Health 2020,17,doi:10.3390/ijerph17103703.
7.Line 133: how many samples were used for bootstrapping?
- Thank you for your question. Brief summaries have been provided for each learning model. In the bootstrap method, 500 replacement resampling were performed from the entire testing dataset. We have added a note to the text (Page 8, lines 184–186).
8.Add and discuss more experimental results: confusion matrices.
- Response: Thank you for pointing this out. The confusion matrix for each learning model has been added to the figures and the Discussion section (Page 10, lines 270–273). The added text is as follows:
As indicated by the confusion matrix results, no DL model mistakenly judged CuTS in the control group. US is minimally invasive, and the study showed the possibility of using the DL models for CuTS screening.
9.Table 1: is the difference between the performance of models statistically significant. Perform statistical testing and present the results (p-values).
- Response: Thank you for your question. The results were evaluated by using a cross-validation method. No significant differences were found for any of factors between the models. The results of a statistical evaluation have been added to the manuscript (Page 8; lines 196–198). The added text is as follows:
The results of cross-validation showed no statistically significant differences among the learning models or between the scores of validation and testing, suggesting that overfitting did not occur.
10.The noise in real-world biomedical images is a well-known problem that reduces the accuracy of diagnostics. You should explore the robustness of the proposed method to noise. You can artificially lower the quality for example by adding noise, and then analyse the drop in performance.
- Response: We agree with your observation. In this study, we experimentally measured multiple patients using the same US device from different angles by tilting, sliding, and rotating the probe under the same conditions. In the future, we plan to verify the accuracy by adding data obtained from different instruments to increase versatility. In this study, we could not change the measurement environment; hence, we have added this information as a study limitation (Page 11, lines 327–334). The added text is as follows:
Finally, in this study, the same US device was used in every phase, and no investigation was conducted regarding changes in the environment, such as the addition of noise. The same image may not be obtained even when looking at the same object because the settings of the US device and the examiner's skill may vary. Although multiple patients were examined experimentally using the same US device from different angles by tilting, sliding, and rotating the probe under the same gain and focus conditions, the accuracy of the measurements should be verified by adding data obtained from different instruments to improve generalizability.
11.For multi-part figures, explain each subfigure in the caption of the figures.
- Response: Thank you for your suggestion. An explanation is provided for each figure.

Round 2
Reviewer 1 Report
Authors answered all the questions clearly so I recommend this article to be accepted to the editor.
Reviewer 2 Report
The authors did all the required revisions and improved the manuscript accordingly. I have no further comments and recommend the article to be accept.